# Nanoscale characterization of the biomolecular corona by cryo-electron microscopy, cryo-electron tomography, and image simulation

Sara Sheibani[1,7 ✉], Kaustuv Basu[2,7], Ali Farnudi [3], Aliakbar Ashkarran[4], Muneyoshi Ichikawa[1,5,6], John F. Presley[1], Khanh Huy Bui [1,2], Mohammad Reza Ejtehadi [3 ✉], Hojatollah Vali[1,2] & Morteza Mahmoudi[4 ✉]

The biological identity of nanoparticles (NPs) is established by their interactions with a wide range of biomolecules around their surfaces after exposure to biological media. Understanding the true nature of the biomolecular corona (BC) in its native state is, therefore, essential for its safe and efficient application in clinical settings. The fundamental challenge is to visualize the biomolecules within the corona and their relationship/association to the surface of the NPs. Using a synergistic application of cryo-electron microscopy, cryo-electron tomography, and three-dimensional reconstruction, we revealed the unique morphological details of the biomolecules and their distribution/association with the surface of polystyrene NPs at a nanoscale resolution. The analysis of the BC at a single NP level and its variability among NPs in the same sample, and the discovery of the presence of nonspecific biomolecules in plasma residues, enable more precise characterization of NPs, improving predictions of their safety and efficacies.

[1] Department of Anatomy and Cell Biology, McGill University, Montreal, QC H3A 0C7, Canada. [2] Facility for Electron Microscopy Research, McGill University, Montreal, QC H3A 0C7, Canada. [3] Department of Physics, Sharif University of Technology, Tehran, Iran. [4] Department of Radiology and Precision Health Program, College of Human Medicine, Michigan State University, East Lansing, MI, USA. [5] Department of Systems Biology, Graduate School of Biological Sciences, Nara Institute of Science and Technology, Institute for Research in Fundamental Sciences (IPM), Takayama-cho, Ikoma, Nara, Japan. [6] PRESTO, Japan Science and Technology Agency, Kawaguchi, Saitama, Japan. [7]These authors contributed equally: Sara Sheibani, Kaustuv Basu. ✉email: sara.sheibani@mcgill.ca; ejtehadi@sharif.edu; mahmou22@msu.edu

The biomolecular corona (BC) adsorbed onto the surface of nanoparticles (NPs) after exposure to human plasma controls their interaction with biological systems[1–4]. Among the biomolecules involved in forming the BC (e.g., lipids, nucleic acids, metabolites, and sugars), proteins are the most-studied owing to their function in a wide range of cellular activities and analysis by techniques such as liquid chromatography-tandem mass spectroscopy (LC-MS/MS)[5]. Although LC-MS/MS delivers reliable information on protein identification and quantification, it cannot perceive the arrangement and distribution of the proteins within the BC nor their association and interaction with the surface of NPs[6].

Other techniques to characterize the BC, such as gel electrophoresis, differential centrifugal sedimentation, and conventional electron microscopy, also fail to provide structural details. Alternative approaches, therefore, must be employed to elucidate the structure, distribution, homogeneity, and interaction of biomolecules in the BC and near the surface of the NPs. Complete characterization of all the components of the BC is therefore essential to assess their safety and biological efficacy[6–8].

Biomolecules adsorbed on the surface of NPs can exist as either a soft or hard corona. Kokkinopoulou et al.[9], for example, applied transmission electron microscopy (TEM) and cryo-electron microscopy (cryo-EM) to visualize and discriminate between the soft and hard corona. These authors also showed the BC is an undefined and loose network of proteins. The objective of the current study is to investigate the morphological details of the biomolecules and their distribution and association with the surface of carboxylated polystyrene NPs (PS-COOH NPs) at the nanometer scale using high-resolution transmission electron microscopy (HRTEM), cryo-EM, cryo-electron tomography (cryo-ET), and image analysis.

## Results and discussion

The accumulation and distribution of the biomolecules adsorbed on the surface of PS-COOH NPs after exposure to human plasma were investigated by negative-stain TEM on a Cu TEM grid with carbon support film. The TEM images showed a patchy formation of BC on the PS-COOH NPs (Fig. 1a, b). To distinguish between the soft and hard corona, the PS-COOH NPs were washed three times to remove the loosely-bound biomolecules of the soft corona. The TEM images of this sample showed patchy clusters of biomolecules surrounding the NPs (Fig. 1c, d). While other analytical techniques can assess the quality and quantity of the biomolecules within the soft and hard coronas[1,9,10], TEM can visualize the arrangement and distribution of the proteins in the BC and on the carbon support film between washed and unwashed samples (Fig. 1). Although negative staining can introduce sample preparation artifacts, the uniform distribution of the individual and clusters of biomolecules adsorbed on the surface of the carbon support film of the TEM grids suggests the biomolecules may have flocculated (in the range of a few to 10 nm) in the plasma suspension (Fig. 1a, b). A considerable amount of fine-grained biomolecules remained on the carbon support film even after washing three times (Fig. 1c, d). It was, therefore, surprising to observe the patchy appearance of clusters of biomolecules was more pronounced on the surface of the NPs after washing. It was likely these clusters may have formed before reaching the surface of individual NPs. The formation of the clusters, therefore, has implications for the interpretation of the proteomic data and biological function as well as the biological identity of the NPs, which also may significantly affect the balance between their therapeutic targeting efficacies and safety.

The physicochemical properties of the NPs play an essential role in the composition of the BC[3,11]. Specific proteins in the BC, having distinct sizes and surface charges, suggest a direct interaction (binding) between these proteins and surface of the NPs[3,12]. Despite extensive investigation, however, the mechanisms underlying these interactions remain ambiguous. Our analysis of hundreds of NPs showed a random distribution and concentration of biomolecules adsorbed on the surface of the NPs. Differences in the morphological patterns and appearance of the BC of individual NPs having the same size and surface characteristics from the same experiment suggest a more complex and diverse interaction of biomolecules with the NPs compared to the currently published literature. For example, comparing Fig. 1c and Supplementary Fig. 1 shows the differences in the surface patterns among NPs.

Cryo-EM images of the BC adsorbed onto the surface of monodispersed PS-COOH NPs showed a random distribution of clusters of proteins and other biomolecules adsorbed from the human plasma (Fig. 2). Although these biomolecules are visible within the BC, their association and affinity with the surfaces of NPs are difficult to assess. Some clusters appear to be bound to the surface, whereas others accumulate at distances from the NPs. The cryo-EM images in Fig. 2 are a projection of the original, untreated PS-COOH NPs (Fig. 2a). After incubating the NPs with human plasma, the clusters of biomolecules are projected as darker densities both around and on the surfaces of the NPs (Fig. 2b–d).

As cryofixation preserves the original state of the hydrated biological and colloidal dispersions (see "Methods"), the features observed in the cryo-EM images in Fig. 2 likely represent the initial state of the NPs and their associated BC. The configuration of the NPs and their BCs preserved in the vitreous ice layer of the holey carbon TEM grids reflects the true nature of the BC and their adsorption on the NPs. It was, however, surprising to discover small clusters of proteins and biomolecules around and between the NPs even after washing three times, confirming the existence of plasma residues not associated with the actual biological identity of the NPs (Fig. 2c, d). Although the origin of these residues is unknown, they could induce errors in the analysis of the BC by LC-MS/MS and gel electrophoresis. Most of the proteins associated with these clusters and aggregates do not interact with the surface of the NPs, and therefore are nonspecific and should not be considered part of the hard corona. Analytical results obtained by gel electrophoresis, LC-MS/MS and proteomics incorporate these nonspecific protein residues as part of the protein structure in the BC. The proteins within the clusters adsorbed on the NPs also are dispersed during the preparation of the samples for LC-MS/MS analysis. Therefore, the structure and function of individual proteins trapped in the cluster will be included in the LC-MS/MS and proteomic analyses. Considering the functional groups of the proteins are locked within the clusters and may not be visible in the biological environment, the predicted function of the corona in the proteomic characterization may differ from its actual function under in vitro and in vivo conditions. To assess the origin of the clusters observed around and between the NPs in the cryo-EM images, we froze specimens of pure plasma with the same concentration used in our experiments and a mixture of plasma and NPs with 5× less concentration of NPs. In both samples, the degree of cluster formation was considerably reduced (Supplementary Fig. 2). It is evident at the higher concentration of NPs that are usually used for proteomics analysis (e.g., LC-MS/MS) could result in the flocculation of the protein, leading to the cluster formation in cryo-preparation. Therefore, the use of diluted NPs may significantly reduce the effect of flocculation of biomolecules in corona layer and, therefore, cause less error in proteomics data.

Cryo-ET enabled the acquisition of three-dimensional (3D) images of the proteins and biomolecules within the BC and their

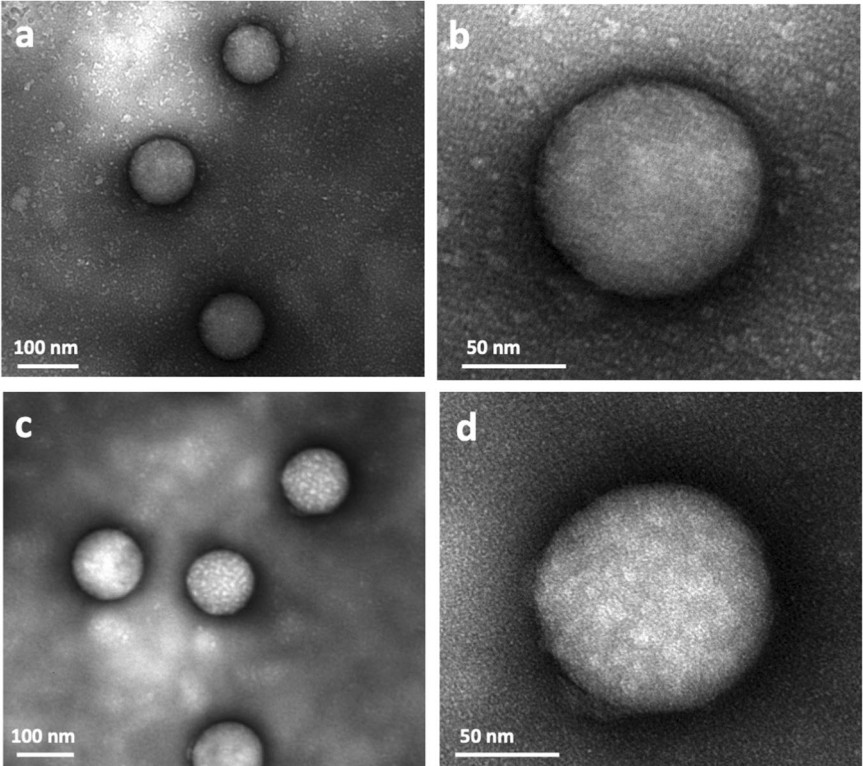

**Fig. 1 TEM images of soft and hard BCs on the surface of NPs.** Clusters of biomolecules are visible on both the carbon support film of the Cu TEM grid and the surface of the NPs before washing (**a**, **b**), and in the hard corona of NPs after washing three times (**c**, **d**). Higher-magnification images (**b**, **d**) of the surface of a single NP shows the clusters of biomolecules in the hard corona. Note the absence of larger clusters in the background of the unwashed sample.

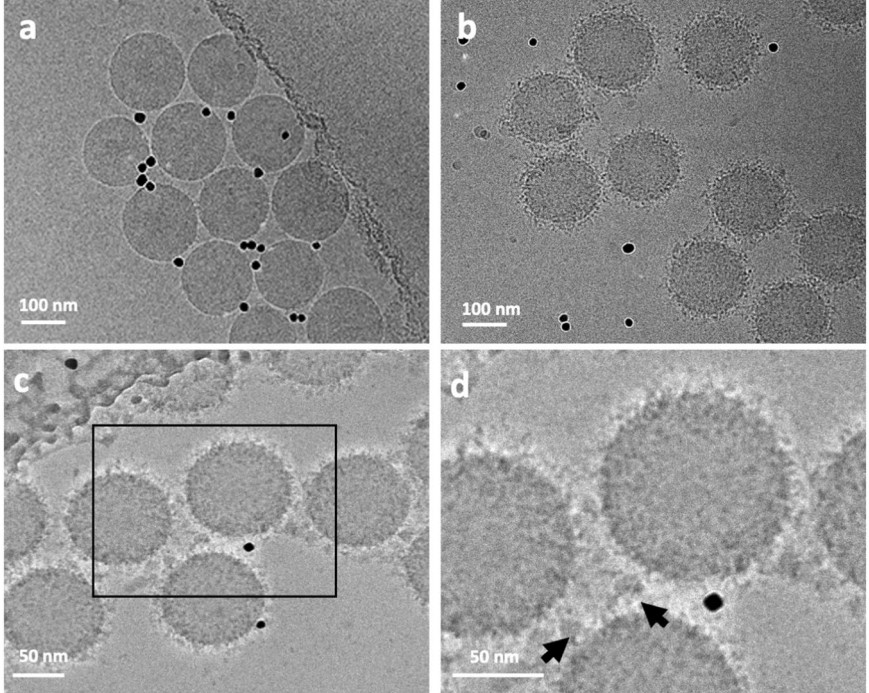

**Fig. 2 Cryo-EM images of the PS-COOH NPs obtained with a direct electron detector and phase plate. a** Original NPs without corona; **b** NPs with corona; **c**, **d** higher magnification of the NPs showing the distribution of biomolecules and their association with the surface of the NPs. Nonspecific proteins accumulate between the NPs as large clusters (black arrows). The black dots in the images are 10 nm gold fiducial markers.

relationship to the NPs. A tilt series of the sample described above, consisting of 2D projection images, was used to reconstruct the 3D volumes of the whole NPs. The reconstructed volumes (Fig. 3 and Supplementary Movies 1–3) showed the random and patchy accumulation of relatively large clusters of biomolecules. Owing to shifts, rotations, and other distortions among the individual 2D projections of the tilt series and the threshold applied in the tomographic reconstruction of the 3D volume, individual molecules within the corona could not be resolved (Fig. 3). Furthermore, the density and size of the clusters and their distribution pattern on the surface of the NPs depend on the threshold selected for image processing.

It should be noted that the threshold selected for the 3D reconstruction can show only larger molecules and clusters of proteins; smaller proteins are better visualized in negatively stained samples (Fig. 1) and individual slices of the 3D tomographic volume (Fig. 4). The patchy structure and random distribution of clusters of proteins on the surface of the NPs, as well as the accumulation of nonspecific proteins and biomolecules between the NPs, are more evident in the cryo-EM images. It is important to emphasize that both the plasma proteins within the BC and those associated with the residues are included in LC-MS/MS analysis, which can be a significant source of error in BC proteomics analysis. As protein functionality within the BC depends on their position and exposure site[3,5,13], the conformation of the same protein should differ, depending on whether it is bound to the surface of the NP or embedded within a cluster.

To better resolve the local densities of the plasma biomolecules and their association with the surface of the NPs, individual slices of the 3D tomographic volume were analyzed (Fig. 4). As mentioned above, it is difficult to show the structure of individual proteins within the BC by conventional reconstruction. The dark, point-like densities around individual slices of the reconstructed tomographic 3D volume (Fig. 4) could represent individual proteins and smaller clusters. Analyzing the virtual slices of the reconstructed 3D volume from the top to bottom of the BC revealed the local distribution and arrangement of the biomolecules and their association with the surface of the NPs. These observations, along with the image analysis, showed significant differences exist in the distribution of biomolecules and their association with the surface of the NPs. These differences appear within the corona and not in the overall thickness of the BC or with other characteristics such as asphericity and anisotropy (as discussed below in the section on image simulation

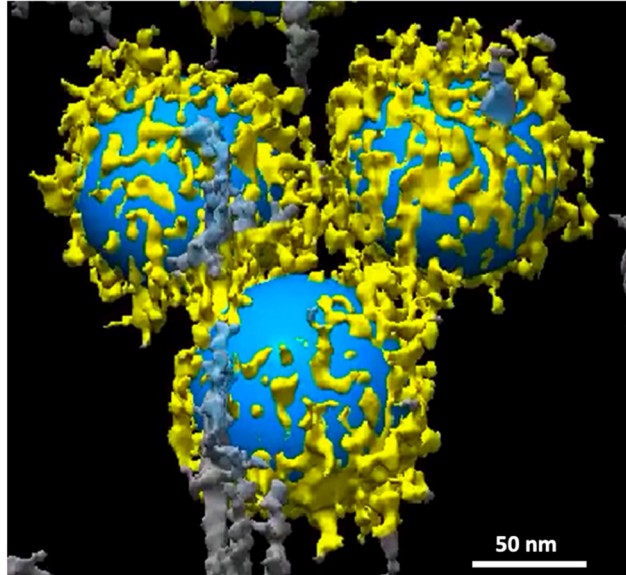

**Fig. 3 Representation of distribution pattern of clusters of biomolecules of the surface of individual NPs.** The image is a snapshot generated from the 3D volume of Supplementary Movie 4.

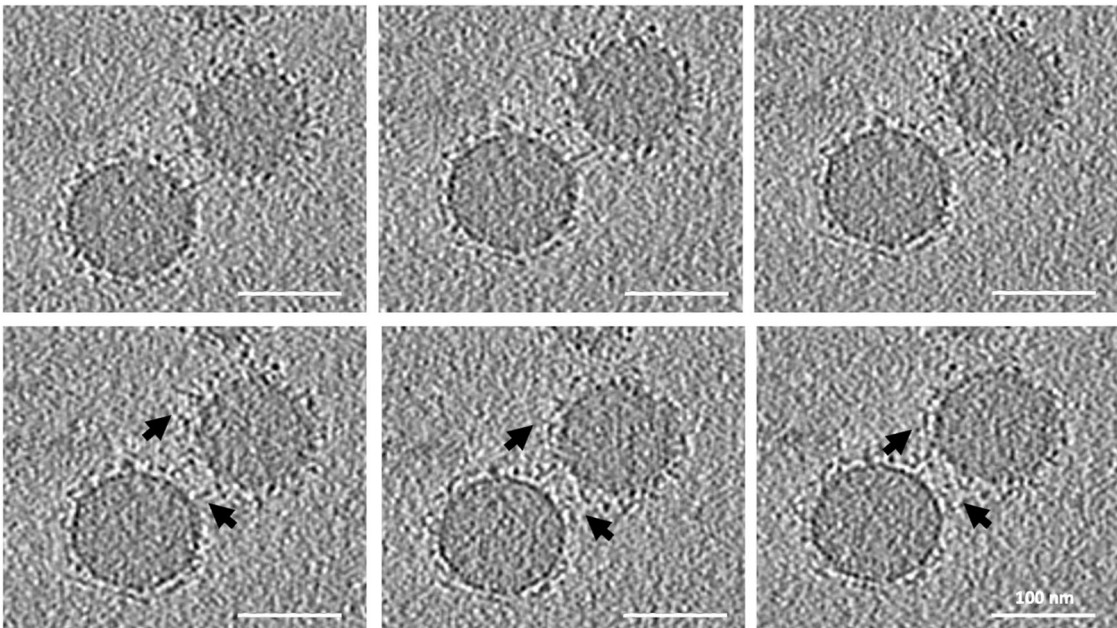

**Fig. 4 Selected slices from the reconstructed tomographic 3D volume (see Supplementary Figs. 3, 4, and Supplementary Movies 5–23 of the Supplementary Information; see the image analysis section for details of the generation of the 3D corona decoration).** The 2D images represent virtual slices through the reconstructed 3D volume of the NPs and corona, from top to bottom. Local distribution and association of the biomolecules within the corona on the surface of the NPs are visible. The biomolecules appear as a dark dense structure bound to the surface of the NPs and at distances from the surface. Changes in the appearance of the individual dots from one image to the next image (see arrows) indicate that the size of these dots, representing biomolecules, is ~1 nm. The scale bar for all images in the panel are the same (100 nm).

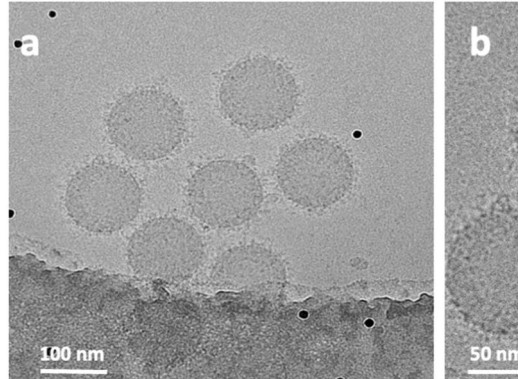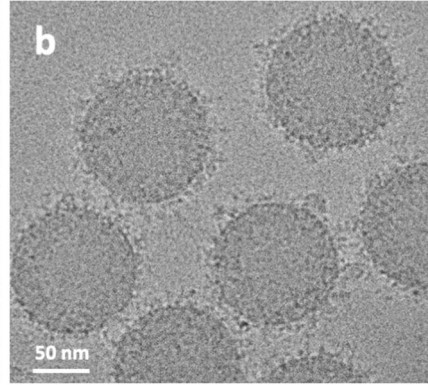

**Fig. 5 Cryo-EM images of amino PS-NPs taken with the Falcon 2 direct electron detector (DED) and phase plate. a** Lower magnification shows of the distribution of the biomolecules, and **b** higher magnification revealing their associations with the surface of the NPs.

model) (see Supplementary Figs. 3, 4, and Supplementary Movies 1–23). Other techniques, such as differential centrifugal sedimentation[10,14] and conventional TEM[10,15], cannot show the distribution of the biomolecules within the corona or their relationship to the NPs. Within the few nanometer-thick-slice of a single NP (see Supplementary Figs. 3, 4, and Supplementary Movies 1–23 of the SI), biomolecules appeared to be both bound to and at distances from the surface of the NPs. This may be due to the (i) protein–protein interactions and (ii) formation of multilayer BCs. Figure 4 includes six consecutive slices from the 80 slices of the whole 3D tomographic volume of an individual BC-coated NP. While each slice is ~1.2-nm thick, the protein organization is different between individual slices within the ~6-nm-thick stack of six slices (arrows in Fig. 4). The complete 3D tomographic volume containing all 256 slices is shown in Supplementary Figs. 2 and 3. Movies of the 3D volume of the BC-coated NPs are provided in Movies 1–23 of the SI.

While traditional negative staining can produce artifacts, cryofixation preserves the original in situ structure. At a cooling rate >$10^5$ K × $s^{-1}$, water is converted to a glass-like structure known as vitreous ice[16]. Under these conditions, no freezing artifacts are introduced, and the original structures and molecular interactions are preserved. Therefore, the heterogeneous distribution of the biomolecules and the diversity in the molecular pattern observed in the slices of the reconstructed tomograms represent the real association of the biomolecules with the surface of the NPs. It is important to emphasize that the patchy pattern observed in the 3D volume shown in Fig. 3 represents only the clusters within the corona. Analysis of the individual slices, however, revealed that the surface of the NPs is covered by smaller molecules not visible in the reconstructed 3D images. Kokkinopoulou et al.[9] focused on the patchy structure of the clusters and missed documenting the presence of the smaller biomolecules.

Using the magnetic levitation technique (which is capable of accurate and robust density evaluations and differences even at the atomic level in chemical compounds/molecules[17–19]), we were able to demonstrate the existence of a heterogeneous BC in the hard corona-coated polystyrene NPs[20]. It should be mentioned the in situ corona-coated NPs (in the presence of excess plasma) could not be evaluated by the magnetic levitation system owing to the instability of excess plasma in the paramagnetic solution[20,21]. The cryo-TEM data presented herein is the first successful attempt to visualize the association of individual biomolecules with the surface of individual NPs at the nanoscale. As a 2D projection, one should keep in mind these molecules are interacting with other biomolecules in the BC not visible in this projection.

To validate the observed BC structures in other types of NPs, we conducted experiments with positively charged amino PS-NPs under the same conditions as the PS-COOH NPs (Fig. 5). Based on analytical data, such as gel electrophoresis, LC-MS/MS/MS, and differential centrifugal sedimentation, a clear difference existed in the total volume of proteins within the corona of negatively and positively charged NPs[3,14,22,23]. Our cryo-EM analysis, however, showed no significant morphological differences between the two types of samples. We also investigated the structure of BC on other types of widely used NPs (e.g., iron oxide and gold). As shown in Fig. 6, a distinct difference appears between the uncoated gold NPs and those bound with BC. Owing to the high atomic mass and strong adsorption of the electron beam of Au NPs, however, it was difficult to distinguish morphological features of the corona.

From the cryo-EM data, we developed a simulation algorithm in C++ to model individual 2D tomographic slices (Fig. 4). The opaque core represents uncoated NPs and the clusters of biomolecules as dots associated with the BC surrounding the NPs. The coordinates of the voxels (volumetric pixels) represent the 3D structure of the BC and were extracted to calculate the geometric properties of the clusters. To determine the voxel dimensions (see Supplementary Figs. 3 and 4 complete tomographic slices), we assumed each image slice had a thickness equal to the interval between the image slices (limit of resolution). Therefore, the voxels are cubes with a side length of 1.2 nm.

The algorithm separates the BC from the background noise by setting a binarization threshold that colors the noise and extra artifacts (black) and the BC and NPs (white) through a series of iterations. Figure 7 is a normalized histogram of the low-concentration sample. The rescaled histogram has a bell shape, where the peak represents the intensity values that are essentially the background noise. If the corona had more contrast, a second peak would have appeared in the histogram, making the binarization threshold easily identifiable. The green dots in Fig. 7 represent the normalized number of counted clusters as a function of the binarization threshold. If the threshold is near the intensity color of the background, the binarized image will contain a small number of very large clusters (Fig. 7, thresholds larger than 120 and less than 80). An acceptable threshold must result in an image that can differentiate between voxels containing the BC and the solution and form a large number of small clusters. Two thresholds were found with such a property, as shown by the red dashed loop in Fig. 7.

We found a threshold range that can differentiate voxels containing coronas in the solution, shown by the red dashed loop. Applying thresholds within the range that included the two bins will yield an image that can be used to extract very similar results.

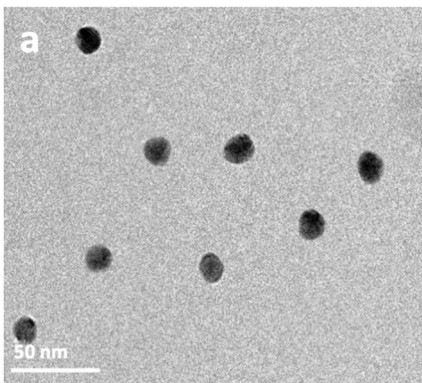
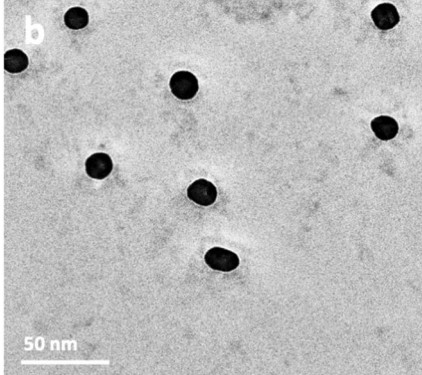

**Fig. 6 Cryo-EM images of 15-nm gold NPs. a** NPs without BC and **b** NPs with BC. Note the occurrence of biomolecules in the plasma residue in the background.

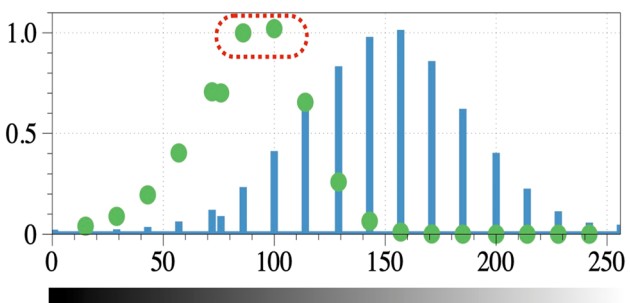

**Fig. 7 A plot of the rescaled histogram of the low-concentration image stack (blue bars) and the rescaled number (normalized) of clusters detected as a function of the binarization threshold (green dots).** The red dashed line shows the grayscale intensity bins that yield the largest number of clusters. Thresholds greater or less than these thresholds create artifacts in the solution owing to background noise. The accuracy in identifying the binarization threshold is highly dependent on the contrast of the images. The horizontal axis shows the grayscale intensity values. Both the histogram and the number of clusters have been normalized to their maximum value to permit comparison of peak locations.

Figure 8 is a representative example of the results after applying the thresholds identified by either of the two bins.

Figure 8a shows the tomographic slice 42 of Supplementary Fig. 3 (as an example) of the low-concentration sample. The dark tone (i.e., high electron density) of the corona makes it distinguishable from the background noise. Figure 8b shows the slice after applying the calculated threshold with the clusters of BC in white. Since the NPs also have a dark tone, the same threshold can be applied to identify them.

We implemented a simple extension of the standard 2D Hough transformation, introduced in 1962[24], to identify spherical shell-like objects (the NPs) in the image stack. Figure 8c shows the 42nd tomographic slice after the Hough transformation, where the center of the NP is the bright point. The NPs closer than 11 nm were removed from the analysis to isolate the BCs affected by one NP. To identify the BC, we used a cluster detection algorithm to search for neighboring voxels that make a cluster with more than four voxels. To include the BC in contact with the surface of the NPs in the analysis, we manually removed the voxels that build the NPs. Finally, we produced isolated 3D images of all NPs [see Supplementary Movies 5–14 of Supplementary Fig. 3 (for 10% plasma concentration) and Supplementary Movies 15–22 of Supplementary Fig. 4 (for 50% plasma concentration) for details]

to understand better the 3D structure of the BC surrounding the NPs.

Figure 9a is an image containing NPs, BC, and other artifacts. In Fig. 9b, we isolated and binarized a single NP (blue) and the surrounding proteins (white). The coordinates of the center of mass of each cluster were used to calculate the distance from the surface of the NP (Fig. 9c).

The gyration tensor of each cluster was calculated (i) to analyze its structural features and (ii) identify the existence of dispersion of proteins near the NPs, which is defined as,

$$S = \frac{1}{N} \begin{pmatrix} \sum_i (x_i - x_{com})^2 & \sum_i (x_i - x_{com})(y_i - y_{com}) & \sum_i (x_i - x_{com})(z_i - z_{com}) \\ \sum_i (x_i - x_{com})(y_i - y_{com}) & \sum_i (y_i - y_{com})^2 & \sum_i (z_i - z_{com})(y_i - y_{com}) \\ \sum_i (x_i - x_{com})(z_i - z_{com}) & \sum_i (z_i - z_{com})(y_i - y_{com}) & \sum_i (z_i - z_{com})^2 \end{pmatrix}$$

(1)

where $N$ is the total number of voxels in a single cluster, and $x_i$, $y_i$, $z_i$, $x_{com}$, $y_{com}$, and $z_{com}$ are the coordinates of the voxels in a cluster and the coordinates of the center of the cluster, respectively. Although the parameters are invariant under the rotation of the axes, we diagonalized the gyration tensor $S = diag(\lambda_1^2, \lambda_2^2, \lambda_3^2)$ and sorted the eigenvalues in descending order $\lambda_1 \geq \lambda_2 \geq \lambda_3$ to simplify the presented results. The trace of the gyration tensor is the square of the radius of gyration.

$$Tr(S) = \lambda_1^2 + \lambda_2^2 + \lambda_3^2 = R_g^2$$

(2)

Because the radius of gyration is a measure of the radius of the cluster if it had been replaced by a hypothetical sphere, it is an acceptable estimate of cluster size. Figure 9d illustrates a 2D representation of a cluster fitted with a circle with the radius of gyration.

The eigenvalues of the gyration tensor were interpreted as the radii of an ellipsoid that takes up the same space as the cluster (Fig. 9e). Asphericity measures the average deviation of the ellipsoid from spherical symmetry and is defined by subtracting the mean of the two smaller radii $\lambda_2^2$ and $\lambda_3^2$ from the larger one. If the cluster forms a perfect sphere, the asphericity will be zero.

$$b = \lambda_1^2 - \frac{1}{2}(\lambda_2^2 + \lambda_3^2)$$

(3)

Figure 10a is a plot of the average diameter of the clusters ($2 \times R_G$) on the surface of the NPs. The outcomes reveal the sizes of the NPs in the first 3 nm of both the high- and low-concentration samples. The size of the clusters diverges in the range of 3–9 nm from the surface of the NPs, where it appears that the larger clusters formed in the high-concentration solution.

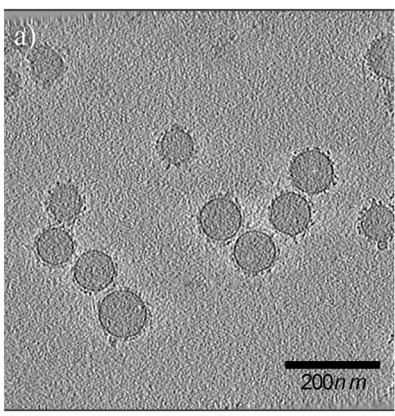
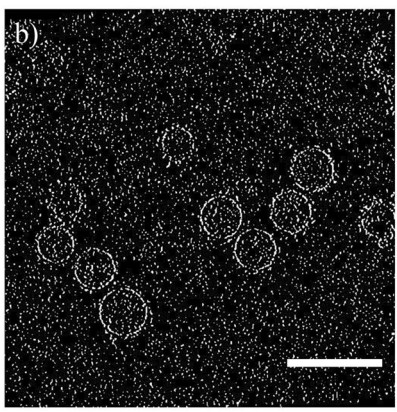
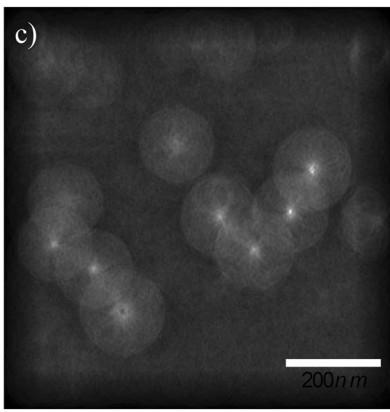

**Fig. 8 Representative example of applying the thresholds, as defined in Fig. 7. a** The 42nd raw slice of the low-concentration sample. **b** The slice after implementing the optimum binarization threshold. **c** The center of the NPs, as detected via the Hough transformation, extended to 3D spherical shells.

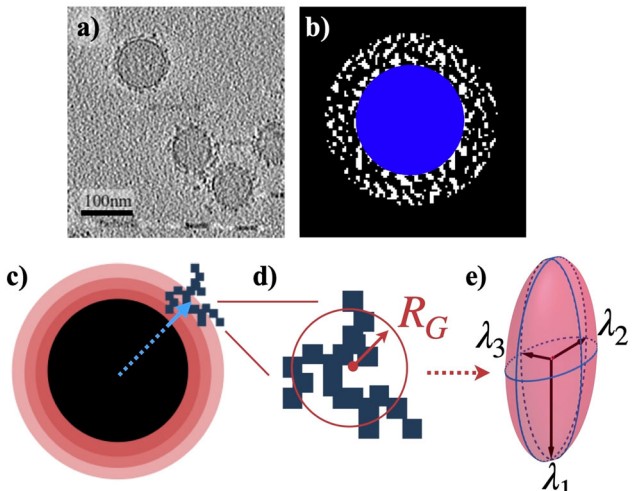

**Fig. 9 Defining the physical characteristics of biomolecular corona.**
**a** A tomographic slice shows the BC surrounding the NPs (full series of 256 distinct cryo-EM images are provided in 10%Corona.rar and 50%Corona. rar files of the SI). The large circles represent spherical NPs. The BC is represented by black points forming clusters around the NPs. The horizontal black bar-like shapes in the bottom right corner of the image are artifacts of smaller gold NPs (used for focusing purposes) that are used to measure the scale of the image. **b** A binarized image showing the BC (white) surrounding the NP (blue). **c** The space around the NP is divided into spherical shells. The blue arrow points to the center of the cluster. The position of each cluster was determined according to where its center is located among the various shells. **d** A schematic representation of an arbitrary cluster. The size of the cluster was characterized by a sphere with a radius of gyration $R_G$. Here we show a 2D representation. **e** The eigenvalues of the gyration tensor are represented with the radii of a hypothetical ellipsoid that best fits the cluster.

The difference in the size of the corona between the two samples is negligible beyond a distance of ~9 nm. Figure 10b shows a plot of asphericity for both samples. While the samples showed similar properties in the first 3 nm, the clusters in the 50% plasma concentration are stretched out compared to the 10% plasma concentration in the 3–9 nm space around the NPs. Beyond this region, we saw a slow transition to a more spherical structure, as expected in an isotropic environment. Finally, we calculated the anisotropy of the gyration tensor in Fig. 10c. As anisotropy reflects both the symmetry and the dimensionality of a cluster, it was calculated as follows (values are limited to the

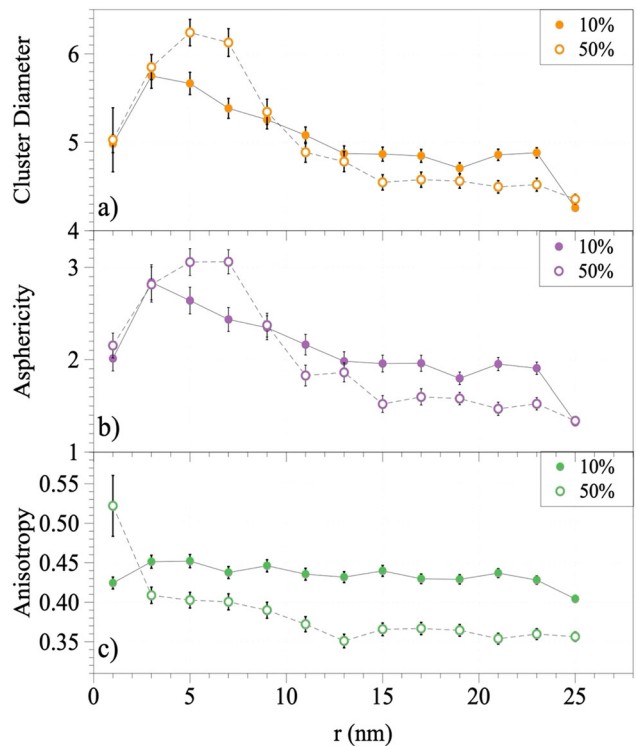

**Fig. 10 Detailed physical properties of the corona clusters at the surface of polystyrene NPs.** Plots of **a** the average cluster diameter distribution, **b** asphericity, and **c** anisotropy calculated via the eigenvalues of the gyration tensors of the corona clusters for the 10% (filled circles) and 50% (hollow circles) plasma concentration samples. The horizontal axis shows the distance of the clusters from the surface of the NPs. Error bars = standard deviation ($n = 12$ for the 10% concentration, and $n = 9$ for the 50% plasma concentration).

range from 0 to 1).

$$\kappa^2 = 1 - 3\frac{\lambda_1^2\lambda_2^2 + \lambda_2^2\lambda_3^2 + \lambda_3^2\lambda_1^2}{\left(\lambda_1^2 + \lambda_2^2 + \lambda_3^2\right)^2} \tag{4}$$

It reaches 1 for the limiting case of a rod-like cluster (linear chain in case of polymers) and drops to zero for symmetric conformations (a sphere). Anisotropy converges to 0.25 for objects with planar symmetry[25,26]. The protein clusters in the 50% plasma concentration are more symmetrical compared to the 10% plasma concentration, as shown in Fig. 10c. The symmetry

of the BC formation was affected by the presence of the NPs, and very close to the surface, the anisotropy of the 10% plasma concentration was larger than the 50% concentration. Since, on average, they are the same size (Fig. 10a), the stretching might be due to the surface effects of the NPs. As the corona on the surface of the NPs may be more densely packed, they sometimes appear as very large clusters and cannot contribute to the structural information.

The calculated thickness of BC is, therefore, in good agreement with the published literature applying the technique of differential scanning calorimetry (DSC)[10,14]. Cryo-EM, cryo-ET, and image analysis may address the major shortcomings of the DCS in estimating protein density[21].

In conclusion, the scientific and technological challenges in designing and developing nanomaterials for diagnostic and therapeutic applications are described in recent publications by members of our team[6,27] and others[28]. One of the main challenges is our poor understanding of the mechanism(s) of protein–protein interaction within the corona and the relationship and association of biomolecules with the surface of NPs. The application of cryo-EM, cryo-ET, 3D reconstruction and image processing, and image simulation showed the variation in the structure and distribution of the biomolecules in the coronas of hundreds of individual NPs. It also enables the visualization of the individual biomolecules associated with the surface of NPs by analyzing the nanosized slices of the 3D tomographic volume of a single NP.

The detailed morphological analysis of the BC shows analytical techniques, such as LC-MS/MS and gel electrophoresis analysis, might not represent the true composition of the proteins associated with the hard corona as they average information from billions of NPs. Furthermore, our findings could lead to a better understanding of the function of the BC and its nonuniform behavior, at a resolution of a single NP, which significantly affects the outcomes of in vitro and in vivo experiments as well as most of the reported clinical trials.

An outcome of this study for LC-MS/MS analysis and proteomics is the discovery of a significant amount of small, agglomerated residual biomolecules (~<10 nm) trapped within the BC layer or clustered between NPs and not associated with the BC. Since the same washing protocol was used in both LC-MS/MS and cryo-EM, these residues are included in the pool of proteins removed from the soft coronas.

The other crucial aspect of the current study that introduces further implications for the function of the BC is imaging the distribution of the biomolecules associated with the corona and their relationship with individual NPs at the scale of the single NP. Comparing the morphological characteristics of hundreds of single NPs in the same sample, it is evident that the interaction of biomolecules with the surface of NPs is heterogeneous. Both negative-stain TEM and cryo-EM reveal the biomolecules associated with the BC display a patchy structure of clusters with a heterogeneous distribution over the surface of NPs. It is important to emphasize, however, that these patterns only represent the larger molecules and clusters associated with the BC and not the smaller molecules/proteins covering the entire surface of the NPs as demonstrated in Fig. 4.

Although we made an extensive effort to provide the most reliable analysis of the 3D structure of the biomolecular corona, owing to limitations in currently available technical and experimental capabilities, the results may have subtle variations with the original native structure of the biomolecular corona. We performed several experiments under a variety of conditions to minimize technical problems associated with sample preparation and imaging techniques. Despite the most advanced imaging and analytical approaches, the results of the same experiment on the identical material under the same conditions show considerable variation in the arrangement and distribution of the biomolecules within the BC. The observed dissimilarity of biomolecules within the BC is in good agreement with the second law of thermodynamics, as the system containing dissimilar BC compositions would have much more probable options (i.e., higher entropy) compared to the uniform BC (i.e., lower entropy) across the surface of all NPs within the sample.

In contrast to analytical techniques, such as DSC, that provide semi-quantitative analysis limited to the thickness of BC, the combination of cryo-EM, cryo-ET, and image analysis enabled the detailed physical characteristics of the corona (i.e., dimensions of the molecular clusters, anisotropy, asphericity distribution, and eigenvalues of the gyration tensor). Our findings demonstrate that the application of therapeutic NPs is more challenging than predicted in the published literature. Biosystems, including the immune system, respond to NPs at the level of a single NP. The heterogeneous nature of the BC, therefore, significantly affects their safety and biological efficacy. Last but not least, our findings also suggest that the employed combination of the electron microscopy techniques and image analysis can be used as a gold standard for defining (i) the purity and (ii) homogeneity/heterogeneity of the BC at nanoscale resolution. The former helps nanomedicine community to define the accuracy and reliability of proteomics and analytical chemistry data of the BC at the surface of NPs; the latter is important for defining the suitability of various types of NPs for clinical applications. This means that NPs with a high degree of heterogeneity, should not be used in human applications as those individual NPs across the same sample will have significantly different biological fate, pharmacokinetic, and toxicity.

## Methods

**Materials**. Functionalized (carboxyl and amino) polystyrene nanoparticles were purchased from Spherotech Inc. (www.spherotech.com).

**BC sample preparation**. Buffer-diluted polystyrene NPs were mixed with human plasma (Innovative Research) separately to a final concentration of 10% and 50%, respectively. The concentration of polystyrene NPs (0.62 mg/ml) was the same in both plasma dilutions (total volume: 500 μl) were incubated at 37 °C for 1 h under constant agitation at 10 G. Subsequently, they were centrifuged at $21,952 \times g$ for 30 min at 10–15 °C to pellet the polystyrene NPs–protein complexes. To remove the soft BC, pellets were resuspended with 400 μL of cold Sorensen's phosphate buffer and centrifuged again for 10 min. To reduce contamination of proteins with low affinity to the PS-NP surface (the soft BC), the pellets were washed three times and resuspended in 200 μl Sorensen's phosphate buffer. Structural and physical characterization of the hard BC complexes was performed.

**Negative-stain TEM and cryo-EM**. Optimal concentration of PS-NPs containing BC was determined by negative staining using 2% uranyl acetate. Aqueous suspensions (5 μL) of the prepared samples were dispersed and pipetted onto a 200-mesh copper TEM grid with carbon support film and allowed to dry under at room temperature. The samples were imaged with a FEI Tecnai G2 F20 200 kV TEM equipped with a Gatan Ultrascan 4000 CCD Camera Model 895.

For each sample prepared for cryo-EM, 10-nm BSA-treated gold NPs were mixed with the optimal concentration of PS-NPs containing BC at a ratio of 1:4.3. Five microliters of the sample was applied to a glow-discharged holey carbon grid (C-Flat R2/2, Protochips, Inc), blotted, and frozen in liquid ethane using the Vitrobot Mark IV (Thermo Fischer Scientific, Hillsboro, OR, USA). Images were collected using a Titan Krios 300 kV Cryo-S/TEM equipped with a Falcon 2 direct electron detector (DED and phase plate (Thermo Fischer Scientific, Hillsboro, OR, USA) at a nominal magnification of 75k× having a pixel size of 1.075 Å, with defocus ranging from −2.0 to −3.0 μm under low-dose conditions.

**Cryo-electron tomography**. Single-axis tomograms were collected by a Titan Krios TEM (Thermo Fischer Scientific) operated at 300 kV and equipped with Falcon 2 DED) (Thermo Fisher Scientific) using FEI Batch Tomography Software. The cryo-tilt series was collected at a magnification of 59k× over a tilt range of ±60° with an angular increment of 2°. The nominal pixel size was 1.375 Å, the defocus ranged from −2 to 3 μm, and the total dose per tomogram was ~80 e−/Å².

**Image processing**. All tomograms were aligned, filtered, and reconstructed using IMOD.4.9.11. The back-projection method and iterative reconstruction techniques (SIRT) were used. The 3D visualization and volume measurement were performed and movies were generated with UCSF Chimera software version 1.5.1[29–31]. The movies 4–32 were generated using ImageJ software version 2.0.0-rc-69/1.52p.

**Statistics and reproducibility**. For negative stain and cryo-EM, multiple TEM grids were prepared under different concentrations and different staining or freezing conditions. Each experiment was repeated two to three times. The representative images included in the paper were selected from a set of 30 to 40 images. For the cryo-electron tomography, three different grids from each sample were prepared and run in TEM for collecting data for 3D reconstruction.

**Reporting summary**. Further information on research design is available in the Nature Research Reporting Summary linked to this article.

## Data availability
All relevant data (including the raw data of electron microscopies) are available from the authors.

## Code availability
The imaging codes used in this study are available through the following link: https://github.com/afarnudi/Corona_Structure_anlysis.

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

## Acknowledgements
We would like to thank members of the Facility for Electron Microscopy Research at McGill University, Mrs Jeannie Mui and Dr. S. Kelly Sears for their support.

## Author contributions
Conceptualization: S.S., H.V., and M.M.; methodology, experimental, and Image analysis: S.S., K.B., A.F., A.A., M.I., J.F.P., K.H.B., and M.R.E.; writing—original draft preparation: S.S., A.F., H.V., M.R.E., and M.M.; writing—review and editing: S.S., H.V., and M.M.; supervision: S.S., H.V., M.R.E., and M.M.

## Competing interests
Dr. Mahmoudi discloses that he receives royalties/honoraria for his published books, plenary lectures, and licensed patent (to Seer Inc).
