## [Peer Review File · Nature Communications]

Reviewers' comments:

Reviewer #1 (Remarks to the Author):

In this work, Sheibani et al. described their efforts of employing cryo-TEM for exploiting biomolecular corona. By combined use of 3D reconstruction, simulation and quantitative analysis, authors further investigated the structure of the protein corona. Particularly, they identified the existence of protein clusters within the corona and heterogeneity nature on a particle-by-particle level. The topic of nanoparticle-biomolecule interaction is certainly an important issue worthy deep understanding. The current work makes an impressive step forward in better visualizing the nearly native status of protein corona by using advanced cryo-TEM techniques. The whole manuscript is generally well organized. Publication in Nat. Commun. can be considered upon proper revision:

Authors used multiple centrifugation to prepare nanoparticles with "hard corona only", which should consider the possible effect of centrifugation on the nature of protein corona. Is it possible the observed protein clusters on the corona partly due to the centrifugation processing?

Line 142, "...that these aggregates are not included in the gel electrophoresis". Readers may be curious why authors make this claim. If protein aggregates do exist within the corona, will native gel electrophoresis be able to identify their presence? Relevant data is suggested to be included.

Necessary characterization data of these polystyrene NPs employed in the experiments should be provided. Is the observation from the present cryo-TEM (e.g., protein clusters, heterogeneity) related to the chose nanoparticles or their surface characteristics?

Caption to Fig. 1, b should be the magnification images, which should be corrected. Necessary units should be added for Figure 5.

Line 283, "NPs closer than 11 nm to each other were removed from the 284 analysis to isolate BC that are affected by one NP." The underlying reason to make this choice should be explained in the text.

Biological implication of the observed protein clusters within the protein corona on the surface of nanoparticles needs to be addressed.

Reviewer #2 (Remarks to the Author):

This manuscript reports about the morphology of the biomolecular protein corona of carboxylated polystyrene nanoparticles by Cryo-EM analysis also reporting a 3D-reconstruction of the BC. The manuscript is well-written and the data are well presented. The experimental analysis to extract the structural details of the protein corona seems quite accurate but I am not an expert on the field and I cannot judge about the reliability of these results by the technical point of view. I think that the use of cryo-EM to obtain important details on the morphology, density and homogeneity of the BC structure is undoubtedly a step forward in this field and should become more and more used. Although comparing these results to those reported in the paper Kokkinopoulou, M. et al. "Visualization of the protein

473 corona: towards a biomolecular understanding of nanoparticle-cell-interactions. *Nanoscale* 2017, 9 (25), 8858-8870", cited as reference 9 by the authors, I do not see enough novelty in this study to be published in Nature Comm. I think that this manuscript is valid and it is an important contribution for the research community working on nanomedicine but I believe it should be published in a different journal as a full article and not as a communication.

Reviewer #3 (Remarks to the Author):

In the paper entitled „Studying formation, density, and 3D Structure of Biomolecular Corona by Cryo-Electron Microscopy“ the group of Mahmoudi describes the biomolecular corona and its investigation by electron microscopy. The findings are sound and nearly all are supported by the provided data. The manuscript is well written and scholarly presented, although I would think that Mahmoudi et al. could have written a more interesting paper. The paper is an addition/verification of the literature published before. I would agree that there have not been too many papers published in the field of biomolecular corona and electron microscopy studies. Therefore I would feel that his paper should be published although I would think that it would fit better in another journal.

Here are my comments/suggestions:

It was not easy to find a statement that has a high degree of novelty. The authors state that the individual particles do not look the same in terms of coating with the biomolecular corona. This would be highly interesting. But: I do not agree. I would not expect that each biomolecular corona looks exactly the same, but they look quite similar. Every nanoparticle has proteins/biomolecules on the surface of the nanoparticle and the study cannot distinguish the different proteins/biomolecules. Therefore, I felt that this finding is not supported by the data presented in this aspect.

On page 8 the authors state “It is obvious that these aggregates are not included in the gel electrophoresis and proteomics analysis which, in turn, significantly influences the interpretation of the data.” This depends largely on the preparation methods and there have been methods described to preserve the soft protein corona e.g. AFFF.

The authors also note: “The BC is composed of undefined and heterogeneous molecules.” Although it is correct that this – nor any known study – can pinpoint the exact protein in the biomolecular corona in such images, I would not call these “molecules” undefined. We just cannot define them. There may be methods to come by which we can do exactly so. The authors may state this more clearly.

Figure 1a and 1b (after washing) show aggregated proteins in no relation to the particle surface. The authors have mentioned that they washed once. In the Materials and Methods three washing steps are mentioned. If stated correctly in the main text: This single washing step did not remove the unbound biomolecules/proteins. On the other hand Figure 1d and Figure 2 demonstrates no unbound protein clusters. So was the washing procedure not the same in the whole study? I would therefore feel that further studies are warranted by performing a series of washing steps and how this affects the unbound clusters of biomolecules. This is what is done for most proteomic studies by LC-MS.

Therefore I would conclude that this is a good paper, but I feel that the additional data provided over the status in the field is not overwhelming and that therefore this paper should be published in a lower ranking journal.

Author's Response to Reviewer #1:

In this work, Sheibani et al. described their efforts of employing cryo-TEM for exploiting biomolecular corona. By combined use of 3D reconstruction, simulation and quantitative analysis, authors further investigated the structure of the protein corona. Particularly, they identified the existence of protein clusters within the corona and heterogeneity nature on a particle-by-particle level. The topic of nanoparticle-biomolecule interaction is certainly an important issue worthy deep understanding. The current work makes an impressive step forward in better visualizing the nearly native status of protein corona by using advanced cryo-TEM techniques. The whole manuscript is generally well organized. Publication in Nat. Commun. can be considered upon proper revision:

Authors used multiple centrifugation to prepare nanoparticles with "hard corona only", which should consider the possible effect of centrifugation on the nature of protein corona. Is it possible the observed protein clusters on the corona partly due to the centrifugation processing?

Any sample preparation step under laboratory conditions may produce artifacts, and the procedures involved in the preparation of the corona and its morphological characterization are not an exception. Our data suggest the clusters and aggregates of biomolecules observed on the surface of the NPs are also present in the media before being removed by washing steps. It is difficult to determine whether the clusters seen on the surface of the NPs were deposited on the surface and not removed by the washing. This phenomenon was observed in the three different samples investigated in this study. Without a good understanding of the mechanism involved in the interaction of biomolecules with the surface of any NP, the random patterns observed on the surface of individual NPs suggests nonspecific binding of the biomolecules within the clusters. Note - we followed a preparation procedure employed for decades in both analytical and morphological studies.

Line 142, " ...that these aggregates are not included in the gel electrophoresis". Readers may be curious why authors make this claim. If protein aggregates do exist within the corona, will native gel electrophoresis be able to identify their presence? Relevant data is suggested to be included.

We have clarified this point in the revised version of the manuscript. We distinguished between the cluster/aggregate associated with the NPs, and the nonspecific clusters/aggregates of biomolecules present as residue between and around the NPs. The residues would likely alter the analytical data. Considering the inclusion of the aggregates into the native gel electrophoresis, as long as the nature of these clusters/aggregates is unknown, it is difficult to ascertain if all the components within the cluster/aggregate can dissociate in their original state during the preparation for mass spectroscopy and 2-D electrophoresis.

Necessary characterization data of these polystyrene NPs employed in the experiments

should be provided. Is the observation from the present cryo-TEM (e.g., protein clusters, heterogeneity) related to the chosen nanoparticles or their surface characteristics?

We included new experimental data from two other NPs with different sizes and surface characteristics, and which showed similar results. We characterized the experiments with polystyrene nanoparticles, and provided details from prior publications [e.g., *Nanoscale* 12, 2374-2383 (2020) for density, zeta potentials, and sizes (before and after incubation with plasma proteins)].

Caption to Fig. 1, b should be the magnification images, which should be corrected. Necessary units should be added for Figure 5.

Corrected.

Figure 1 is modified and figure caption revised to address the reviewer concern. Scale has been included in the Figure 4.

Corrected.

Line 283, "NPs closer than 11 nm to each other were removed from the analysis to isolate BC that are affected by one NP." The underlying reason to make this choice should be explained in the text.

The threshold was selected because the identified and reported thickness of protein corona on polystyrene nanoparticles, determined by differential centrifugal sedimentation (DCS), was expected to be approximately between 5-10 nm [e.g., *J. Am. Chem. Soc.* 132, 5761-5768 (2010) and *J. Am. Chem. Soc.* 133, 2525-2534 (2011)]. Therefore, below 11 nm, we may have mixed coronas from various particles.

Biological implication of the observed protein clusters within the protein corona on the surface of nanoparticles needs to be addressed.

In the revised version, this point has been extensively described and discussed. We distinguished between the cluster/aggregate associated with the NPs, and nonspecific clusters/aggregates of biomolecules present between and around the NPs as residues. These residues will certainly alter the analytical data. We discussed the presence of the clusters and their implication for interpreting the analytical data, including mass spectroscopy, proteomics and 2-D gel analysis, in great details. In addition, in our revision, we distinguished between these clusters/aggregates and the smaller biomolecules covering the whole surface of the NPs, which could be directly interacting with the surface of the NPs.

Author's Response to Reviewer #2:

This manuscript reports about the morphology of the biomolecular protein corona of carboxylated polystyrene nanoparticles by Cryo-EM analysis also reporting a 3D-reconstruction of the BC. The manuscript is well-written and the data are well presented. The experimental analysis to extract the structural details of the protein corona seems quite accurate but I am not an expert on the field and I cannot judge about the reliability of these results by the technical point of view. I think that the use of cryo-EM to obtain important details on the morphology, density and homogeneity of the BC structure is undoubtedly a step forward in this field and should become more and more used. Although comparing these results to those reported in the paper Kokkinopoulou, M. et al. "Visualization of the protein 473 corona: towards a biomolecular understanding of nanoparticle-cell-interactions. *Nanoscale* 2017, 9 (25), 8858-8870", cited as reference 9 by the authors, I do not see enough novelty in this study to be published in *Nature Comm*. I think that this manuscript is valid and it is an important contribution for the research community working on nanomedicine but I believe it should be published in a different journal as a full article and not as a communication.

We appreciate the positive comments and fair evaluation of our data and the encouragement to review the literature reporting the application of cryo EM to NPs. By comparing our results with data reported by Kokkinopoulou, et. al., 2017, the reviewer feels there is a lack of novelty. We attempted to rewrite several sections to demonstrate the originality of our work. To our knowledge, Kokkinopoulou, et. al. were the first and last reported study to present cryo-EM images as part of their comprehensive study that also included proteomic and cell-NP interaction. They included low-resolution cryo-EM images in their SI without any explanation. Although their EM images display similar features to what we observed in our samples (e.g. presence of clusters/aggregates and residues of biomolecules), there was no mention of these aspect in their article. The originality of our research is the morphological detail, distribution, and association of the biomolecules with the surface of the individual NPs at the nanometer scale. The morphological analysis of the corona on individual NPs, and the discovery of the biomolecular residues not associated with the biomolecules present in the corona, have significant implication for the interpretation of other analytical techniques, such as mass spectroscopy and electrophoresis 1D/2D gel analysis. It should be emphasized that our application of cryo-EM, cryo-ET, image processing, 3-D reconstruction, and simulation revealed the variation in the structure and distribution of the biomolecules in the coronas of hundreds of individual NPs. It also enabled the visualization of individual biomolecules associated with surface of NPs by analyzing tens of subsequent nanosized slices of the 3-D tomographic volume through a single NP. This represents another new discovery not reported by Kokkinopoulou et. al.

Considering the reviewer's concern about the suitability of our work for publication in *Nature Communication*, we are confident the reviewer will agree the revised manuscript fulfills the requirements for publication. Note - our study will be of interest to a wide range of disciplines, including food sciences, materials chemistry, drug development, and biomedical sciences.

Author's Response to Reviewer #3:

In the paper entitled „Studying formation, density, and 3D Structure of Biomolecular Corona by Cryo-Electron Microscopy” the group of Mahmoudi describes the biomolecular corona and its investigation by electron microscopy. The findings are sound and nearly all are supported by the provided data. The manuscript is well written and scholarly presented, although I would think that Mahmoudi et al. could have written a more interesting paper. The paper is an addition/verification of the literature published before. I would agree that there have not been too many papers published in the field of biomolecular corona and electron microscopy studies. Therefore I would feel that his paper should be published although I would think that it would fit better in another journal.

Here are my comments/suggestions:

We thank the reviewer for their positive feedback and respect the opinion of our work being “an addition/verification of the literature published before”. We hope, however, the reviewer would agree with us that the results and interpretation presented in the revised manuscript are original and novel.

As described above, the most important contribution of the manuscript is visualization of individual biomolecules association with surface of NPs. In our opinion, this is a more accurate characterization of the corona and is complementary to conventional analytical techniques, such as mass spectroscopy and differential centrifugal sedimentation, that averages all the components and thickness of the corona in a sample.

It was not easy to find a statement that has a high degree of novelty. The authors state that the individual particles do not look the same in terms of coating with the biomolecular corona. This would be highly interesting. But: I do not agree. I would not expect that each biomolecular corona looks exactly the same, but they look quite similar. Every nanoparticle has proteins/biomolecules on the surface of the nanoparticle and the study cannot distinguish the different proteins/biomolecules. Therefore, I felt that this finding is not supported by the data presented in this aspect.

We agree with the reviewer that some variation in composition and structure of corona is expected among NPs. Considering the functionalized NPs have the same size and surface properties, it was surprising to observe variation in the morphological pattern in NPs adjacent to each other as shown in some of our images. Since the mechanism of the interaction of biomolecules with the surface of the NPs is unknown, it might be useful to present this observation. To ensure the validity of our observations, we also performed several experiments under a variety of conditions to minimize the technical problems associated with sample preparation and imaging techniques. The results of the same experiment on the same material under the same conditions, however, showed considerable variation in the arrangement and distribution of the biomolecules in the corona. Moreover, this is the first qualitative study to investigate the homogeneity/heterogeneity of the biomolecular corona at the level of a single NP. The

observations are also supported by the results of the magnetic levitation technique that showed the heterogeneous composition of BC [Nanoscale 12, 2374-2383 (2020)] using both density variation of the BC NPs and gel electrophoresis of the extracted magnetically levitated particles.

On page 8 the authors state “It is obvious that these aggregates are not included in the gel electrophoresis and proteomics analysis which, in turn, significantly influences the interpretation of the data.” This depends largely on the preparation methods and there have been methods described to preserve the soft protein corona e.g. AFFF.

We have clarified this point in the revised version of the manuscript. We distinguished between the cluster/aggregate associated with the NPs, and nonspecific clusters/aggregates of biomolecules present between and around the NPs as residues. These residues will certainly alter the analytical data. With respect to inclusion of the aggregate into native gel electrophoresis, again, as long as the nature of these clusters/aggregates is unknown, it would be difficult to know if all the components within the cluster/aggregate can be dissociated in their original state by preparation steps used for the mass spectroscopy and 2D electrophoresis.

The authors also note: “The BC is composed of undefined and heterogeneous molecules.” Although it is correct that this – nor any known study – can pinpoint the exact protein in the biomolecular corona in such images, I would not call these “molecules” undefined. We just cannot define them. There may be methods to come by which we can do exactly so. The authors may state this more clearly.

We agree with the reviewer that the identification of the biomolecules/proteins in the BC cannot be ascertained with imaging techniques. Our objective was to present the best possible morphological and structural features of the BC at the highest possible resolution.

We recently employed isothermal titration calorimetry (ITC) to measure the associated heat changes occurring during the interaction between NPs and a biomolecular source [i.e., fibronectin and human plasma, respectively; Nanoscale 10, 1228-1233 (2018)]. Interestingly, the ITC approach demonstrated subtle exothermic heat adsorption of fibronectin protein with NPs in thermograms; however, and to our surprise, we observed a significant continuous exothermic heat that was generated by interactions of NPs with plasma biomolecules, confirming the BC formation agrees with the second law of thermodynamics. Using the magnetic levitation approach, we could again demonstrate that the composition of BC is heterogeneous [Nanoscale 12, 2374-2383 (2020)]. Both the magnetic levitation and ITC findings are in excellent agreement with the cryo-EM data. All of which confirms the formation of heterogeneous BCs at the surface of NPs.

Figure 1a and 1b (after washing) show aggregated proteins in no relation to the particle surface. The authors have mentioned that they washed once. In the Materials and

Methods three washing steps are mentioned. If stated correctly in the main text: This single washing step did not remove the unbound biomolecules/proteins. On the other hand Figure 1d and Figure 2 demonstrates no unbound protein clusters. So was the washing procedure not the same in the whole study? I would therefore feel that further studies are warranted by performing a series of washing steps and how this affects the unbound clusters of biomolecules. This is what is done for most proteomic studies by LC-MS.

This point was corrected in the revision. We have emphasized that the preparation procedure used in our study was the same used for other analytical techniques.

Therefore, I would conclude this is a good paper, but I feel that the additional data provided over the status in the field is not overwhelming and that therefore this paper should be published in a lower-ranking journal.

Reviewer #1 (Remarks to the Author):

The authors have partly addressed my previous comments. Particularly, I am still concerned about the reliability of the observed "protein cluster/aggregates" that can be partly caused by the sample preparation, even it is a commonly adopted corona preparation procedure as authors claimed. Further evidence is needed.

Reviewer #2 (Remarks to the Author):

The revised version of the manuscript from Mahmoudi et al. is significantly improved. It is well-written and the novelty of the proposed technique to study PC complexes is better highlighted and explained. I think that the manuscript might deserve publication in Nature Comm. after the authors have addressed better the points below:

1) Mahmoudi group has recently presented a quite innovative method to isolate PC complexes (by magnetic levitation) and I think that methodology is less invasive than centrifugation/washing and allows to isolate different composition PC complexes from the same kind of NPs. I think that the present manuscript should report at least for one type of NPs, a cryo-TEM analysis of PC-complexes isolated by this technique in comparison to the PC complexes isolated by conventional centrifugation/washing treatment. In fact, the authors show the presence of these protein clusters associated to HC NPs (or as they say present among NP surfaces but not associated to them). Thus, they speculate that the proteins present in these clusters can alter MS results. I am quite intrigued by the presence of these clusters and I wonder if they derive from the method of purification or if they always present in the biological fluids and they are associated to PC complexes and have a biological function. If they demonstrated that they are instead due to the isolation method, I think their statement would be much more reliable.

- a second point is about their impressive tomography images in which they show a tridimensional image of the associated PC. From those images appear clear that some NP surface "regions" are not coated by the protein. Thus it seems that hard PC is not a thick layer of proteins, but it is quite "soft" and this might have some implications on the ability of functionalized NPs with targeting ligands to exploit their function even in the presence of the PC. Can the author comment a little bit on this aspect, what do they think about it?

Reviewer #3 (Remarks to the Author):

The authors have answered the concerns and as there is a lack of manuscripts and evaluations on the 3D structure of the protein corona I support now publication of this paper in Nature Communications.

Reviewer #1 (Remarks to the Author):

The authors have partly addressed my previous comments. Particularly, I am still concerned about the reliability of the observed “protein cluster/aggregates” that can be partly caused by the sample preparation, even it is a commonly adopted corona preparation procedure as authors claimed. Further evidence is needed.

Author Response: We agree with the reviewer that aggregation can result from sample preparation. Considering flocculation and cluster formation is dependent on concentration, we ran experiments using different concentrations of NPs and plasma. The results of these experiments showed the degree of cluster formation was considerably lower in more diluted samples. It is clear the higher concentration of NPs used in the LC-MS/MS analysis and cryo-preparation used in this study could result in flocculation of biomolecules, leading to cluster formation in the cryo-preparation. We have added the following text to the revised manuscript: “To assess the origin of the clusters observed around and between the NPs in the cryo-EM images, we froze specimens of pure plasma with the same concentration used in our experiments and a mixture of plasma and NPs with 5X less concentration of NPs. In both samples, the degree of cluster formation was considerably reduced (Figure S2). It is evident at the higher concentration used for LC-MS/MS analysis and the cryo-preparation used in this study could result in the flocculation of the protein, leading to the cluster formation in cryo-preparation. Therefore, the use of diluted NPs may significantly reduce the effect of flocculation of biomolecules in corona layer and, therefore, cause less error in proteomics data.”

Figure S2. Cryo-EM images of the PS-COOH NPs (a) original plasma diluted in Sorensen’s phosphate buffer to 50%. There is little evidence for presence of large cluster and aggregate; (b) NPs with corona in 2% solution showing the distribution of biomolecules and their association with the surface of the NPs. Similar to the plasma, there is little evidence for large clusters.

Reviewer #2 (Remarks to the Author):

The revised version of the manuscript from Mahmoudi et al. is significantly improved. It is well-written and the novelty of the proposed technique to study PC complexes is better highlighted and explained. I think that the manuscript might deserve publication in Nature Comm. after the authors have addressed better the points below:

1) Mahmoudi group has recently presented a quite innovative method to isolate PC complexes (by magnetic levitation) and I think that methodology is less invasive than centrifugation/washing and allows to isolate different composition PC complexes from the same kind of NPs. I think that the present manuscript should report at least for one type of NPs, a cryo-TEM analysis of PC-complexes isolated by this technique in comparison to the PC complexes isolated by conventional centrifugation/washing treatment. In fact, the authors show the presence of these protein clusters associated to HC NPs (or as they say present among NP surfaces but not associated to them). Thus, they speculate that the proteins present in these clusters can alter MS results. I am quite intrigued by the presence of these clusters and I wonder if they derive from the method of purification or if they always present in the biological fluids and they are associated to PC complexes and have a biological function. If they demonstrated that they are instead due to the isolation method, I think their statement would much more reliable.

Author Response: We thank the reviewer for the constructive comment regarding our magnetic levitation (MagLev) technique for the analysis of the protein corona (Nanoscale 12 (4), 2020, 2374-2383). However, for the MagLev technique, we also used the standard configuration process. The main reason for performing the washing steps was the excess plasma is not stable in the paramagnetic solution, possibly due to the denaturation of the biomolecules (see details in our other publication: Analytical Chemistry 92 (2), 2020, 1663-1668). Likewise, by using unwashed NPs, we could not follow the same clear corona coated NP collection in the MagLev system. The excess plasma could cause the aggregation of NPs (please see figure 1 below for details).

Although we were able to solve the plasma stability in the MagLev system by replacing paramagnetic solutions with superparamagnetic iron oxide NPs and use the strategy for identifying protein density [Analytical Chemistry 92 (2), 2020, 1663-1668] and disease detection [Advanced Healthcare Materials 9 (5), 2020, 1901608], we could not robustly use it in the collection of unwashed corona-coated NPs mainly due to the interference of the iron oxide NPs with the polystyrene NPs.

Figure 1. Levitation profiles of the bare polystyrene nanoparticles, bare plasma proteins, washed and unwashed biomolecular corona coated polystyrene nanoparticles over the time at 0.2 M concentration of gadovist in MagLev system.

- a second point is about their impressive tomography images in which they show a tridimensional image of the associated PC. From those images appear clear that some NP surface "regions" are not coated by the protein. Thus it seems that hard PC is not a thick layer of proteins, but it is quite "soft" and this might have some implications on the ability of functionalized NPs with targeting ligands to exploit their function even in the presence of the PC. Can the author comment a little bit on this aspect, what do they think about it?

Author Response: Overall, the findings of not having very tight and thick corona shell at the surface of nanoparticles are in agreement with the report by Kokkinopoulou, et. al., 2017. Recent findings on the significance role of antibody attachment to the surface of nanoparticles on the shielding role of protein corona would also indirectly support our findings [Nature Nanotechnology 13 (9), 2018, 862-869]. More specifically, pre-absorption of targeting species on the surface of nanoparticles compared with chemical conjugation, can significantly improve their targeting efficacies due to the i) reduced shielding effect of biomolecular corona and ii) increased availability of the antibody targeting sites. If the corona shell is fully tight and thick, there should not have been significant differences between the two antibody-nanoparticle attachment approaches.

Reviewer #3 (Remarks to the Author):

The authors have answered the concerns and as there is a lack of manuscripts and evaluations on the 3D structure of the protein corona I support now publication of this paper in Nature Communications.

Author Response: We thank the reviewer for the supportive comment.

REVIEWERS' COMMENTS

Reviewer #2 (Remarks to the Author):

I think that the revised version of this manuscript is available for publication in Nature Comm